# Heavy metals in the Arctic: Distribution and enrichment of five metals in Alaskan soils

Clarice R. Perryman[1,2]☉*, Jochen Wirsing[3]☉, Kathryn A. Bennett[1,2]‡,
Owen Brennick[4]‡, Apryl L. Perry[1,2]‡, Nicole Williamson[5,6], Jessica G. Ernakovich[4]

**1** Department of Earth Sciences, University of New Hampshire, Durham, New Hampshire, United States of America, **2** Institute for the Study of Earth, Oceans, and Space, University of New Hampshire, Durham, New Hampshire, United States of America, **3** Sociology Department, University of New Hampshire, Durham, New Hampshire, United States of America, **4** Department of Natural Resources and the Environment, University of New Hampshire, Durham, New Hampshire, United States of America, **5** Environmental Science and Studies Program, Towson University, Towson, Maryland, United States of America, **6** Department of Biological Sciences, Towson University, Towson, Maryland, United States of America

☉ These authors contributed equally to this work.
‡ These authors also contributed equally to this work.
* crp1006@wildcats.unh.edu

**Data Availability Statement:** The raw data files used in this analysis can be accessed through the United States Geological Survey (https://doi.org/10.3133/ds759). We used a subset of the Alaskan

## Abstract

Metal contamination of food and water resources is a known public health issue in Arctic and sub-Arctic communities due to the proximity of many communities to mining and drilling sites. In addition, permafrost thaw may release heavy metals sequestered in previously frozen soils, potentially contaminating food and water resources by increasing the concentration of metals in freshwater, plants, and wildlife. Here we assess the enrichment of selected heavy metals in Alaskan soils by synthesizing publicly available data of soil metal concentrations. We analyzed data of soil concentrations of arsenic, chromium, mercury, nickel, and lead from over 1,000 samples available through the USGS Alaskan Geochemical Database to evaluate 1) the spatial distribution of sampling locations for soil metal analysis, 2) metal concentrations in soils from different land cover types and depths, and 3) the occurrence of soils in Alaska with elevated metal concentrations relative to other soils. We found substantial clustering of sample sites in the southwestern portion of Alaska in discontinuous and sporadic permafrost, while the continuous permafrost zone in Northern Alaska and the more populous Interior are severely understudied. Metal concentration varied by land cover type but lacked consistent patterns. Concentrations of chromium, mercury, and lead were higher in soils below 10 cm depth, however these deeper soils are under-sampled. Arsenic, chromium, mercury, nickel and lead concentrations exceeded average values for US soils by one standard deviation or more in 3.7% to 18.7% of the samples in this dataset. Our analysis highlights critical gaps that impede understanding of how heavy metals in thawing permafrost soils may become mobilized and increase exposure risk for Arctic communities.

Geochemical Database Version 2.0 "Best Value Soil" dataset in our analysis. All data used in our analysis are within the paper and its Supporting Information files.

**Funding:** The author(s) received no specific funding for this work.

**Competing interests:** The authors have declared that no competing interests exist.

## Introduction

The Arctic is warming at double the rate of other areas of the globe [1]. Increasing temperatures are driving numerous ecological, cryospheric, and hydrological changes in the northern latitudes. One notable consequence of rising temperatures in the Arctic and sub-Arctic is widespread thaw of permafrost [2,3]. Permafrost thaw not only exacerbates the decomposition and liberation of carbon (C) to the atmosphere [4] and adjacent water bodies [5], but heavy metals stored in permafrost can also be released by thaw and transported into surface waters [6,7]. Heavy metals are present in Arctic soils due to weathering [8], atmospheric deposition [9], and anthropogenic activities including mining and/or smelting [10,11]. In permafrost-affected landscapes, metals may accumulate in frozen soils. As warming and permafrost thaw continue, the potential risk to human health posed by the liberation of heavy metals into surface waters and ecosystems may intensify.

Previous work demonstrates the potential of permafrost thaw as a driver of heavy metal liberation, which depends on each element's solubility and redox chemistry as well as how thaw alters the hydrology of an affected area. Schuster et al. [12] showed that northern permafrost soils store more mercury (Hg) than all other soils, the atmosphere, and the oceans combined. Elevated Hg concentrations have been found downstream of thaw slumps [7], illustrating the potential for permafrost thaw to pose added danger to human health as this Hg may infiltrate water and food resources. Other metals may be similarly released by permafrost thaw. High concentrations of iron, manganese, nickel, and zinc have been found in permafrost-affected soils in northern Siberia [13], and elevated concentrations of iron, aluminum, chromium, and lead have been observed in areas that experienced rapid thaw [6]. Additionally, increases in soil organic matter (SOM) degradation during permafrost thaw can impact metal mobility as metals bound to SOM are released [14]. Plants also take up metals from soil, and do so more rapidly under warming [15] and elevated $CO_2$ conditions [16,17]. While the ability of permafrost thaw to mobilize heavy metals is increasingly recognized, further work is needed to assess the presence of heavy metals in Arctic soils due to their potential to impact human health.

Subsistence hunting, fishing, and foraging are practices that are both nutritionally and culturally important to Arctic peoples; increasing levels of heavy metals in Arctic landscapes, due to liberation from thawing permafrost, would threaten the safety of this practice. Heavy metals and other toxins have been identified in Arctic freshwater ecosystems [18–20], wildlife [21], indigenous foods [22], and indigenous peoples [23,24]. Kenny et al. [25] report that country foods—especially caribou, beluga, ringed seal, and fish—are an important source of protein and vitamins for communities. Due to bioaccumulation, these species contain increased levels of arsenic, mercury, cadmium, lead, and selenium [22] and persistent organic pollutants [26] which have been found in blood and breast milk in communities that consume country foods. Drinking water resources may also be threatened by metal liberation through permafrost thaw, given previous observations of thaw-driven increases in freshwater metal concentration [7,27] and widespread reliance on surface water resources throughout the Arctic [28].

We focus on five heavy metals that have both documented occurrences in Arctic food and water resources and have substantial human health effects: arsenic (As), chromium (Cr), mercury (Hg), nickel (Ni), and lead (Pb). Arsenic is a carcinogen that can accumulate within the body [29] and has been found in concentrations 14 times higher than the safe threshold levels in some commonly consumed Alaskan fish [22]. Lead and mercury cause neurological damage that is most significant in young children and passes easily from nursing mothers to infants [30,31]. Nickel and chromium are both carcinogens and allergens, and chromium can have negative impacts on kidney health [32,33]. The presence of these metals in the Arctic food chain and their subsequent human health effects are well documented, but the amount of heavy metals stored in Arctic soils

and the risk posed by potential metal liberation through permafrost thaw remains largely unknown. While previous work has examined the concentration of heavy metals in soils in sites within the permafrost zone, to date there has been a lack of data synthesizing the prevalence of heavy metals in permafrost-affected soils, and therefore the potential pool of heavy metals that could be released as permafrost thaws on a global scale. Furthermore, studies that quantify the concentration and/or thaw-driven liberation of heavy metals in Arctic soils are generally conducted in remote areas [7,13], so it is unclear if available data on soil metals includes areas in proximity to communities that may be affected by the release of these metals through permafrost thaw.

Our original aim was to conduct a meta-analysis bringing together publicly available, machine readable data to assess the occurrence of heavy metals in soils across the pan-Arctic, but a lack of such datasets in public repositories precluded this analysis. As such, in this study we synthesize available data on the concentration of As, Cr, Hg, Ni, and Pb in soils in Alaska between 60–70˚N, where Arctic tundra is prevalent, to assess the occurrence of elevated metal concentrations and the risk of heavy metal exposure in permafrost-affected soils in this region. The objectives of this study were to (a) to identify the spatial distribution of sites across the Alaskan Arctic where soil concentrations of As, Cr, Hg, Ni, and Pb have been quantified; (b) determine the heterogeneity of soil heavy metal concentrations across land cover types, depth in the soil profile, and in different permafrost zones; and (c) compare soil metal concentrations from Alaska to average concentrations found in other soils in the United States.

## Materials and methods

### Database search

We searched the Arctic Data Center, National Ice and Snow Data Center, and the United States Geological Survey (USGS) databases for datasets containing data for soil concentrations of As, Cr, Hg, Ni, and Pb. The Arctic Data Center and the NSIDC repositories did not contain any databases that met our search criteria (e.g., soil metal concentrations). The USGS had three datasets matching our search criteria: Alaskan Geochemical Database Version 2.0, the PLUTO Soil Database, and the National Geochemical Survey Database. The Alaskan Geochemical Database Version 2.0 (AGDB2), superseded the PLUTO Soil Database and the National Geochemical Survey Database, so only the AGDB2 database was relevant to our query. Our subsequent analysis was performed using the AGDB2 "Best Value" soils data https://pubs.usgs.gov/ds/759/[34].

### Data preparation and analysis

All data cleaning, analysis, and figure creation were performed using R (v. 3.5.2) [35]. Maps were created using ArcGIS (v. 10.7.1). Data cleaning was performed using the tidyverse R package [36] unless otherwise specified. The data were filtered to retain information on: concentrations of As, Cr, Hg, Ni, and Pb, sample depth, location (latitude, longitude), source (land cover type), collection year, and analytical method. The AGDB2 "Best Value Soil" dataset we used in our analysis was pre-screened by USGS chemists for quality control. The USGS Best Value ranking system selected the method providing the most accurate value for each sample that had been analyzed using more than one analytical method. Samples missing latitude, longitude, depth, and land cover type information were removed. Rows containing negative and/or absent values for As, Cr, Hg, Ni, and Pb were also excluded from the analysis. In the instances where groups of samples had identical concentration values, these data were removed as they were assumed to be erroneous. The resulting data table contained 1,151 rows representing individual samples; however, not every sample had a value for the concentration of all 5 metals. Sampling depths were converted into centimeters (cm) and manually binned

into three depth groupings (0–10 cm, 10–30 cm, and > 30 cm) in Excel. Land cover type had been coded by individual research projects over the course of data collection (c. 1995–2010) at USGS. As such, classifications were not consistent across data sources or between samples in close proximity, varying from information on land cover type to drill sample type. To simplify these labels, the classifications describing similar land cover types were grouped together to match broader classifications from recent land-cover maps of Alaska [37,38] in accordance with their location within the state. For example, all previous mine categories were condensed into a single "mine-impacted" group. The resulting land cover type codes are: forested, glacial deposit, tundra, wetland, and mine-impacted. The post-processing data included in our analysis can be found in the S1 File.

Metal concentrations were checked for normality and homogeneity of variance within land cover types and depths using the Shapiro-Wilks Test and Bartlett Test. Both raw and log-transformed data failed to meet the assumptions for ANOVA. The Kruskal-Wallis Test from the agricolae R package [39], which uses a Bonferroni correction and the Fisher's Least Significant Difference post-hoc test was used to assess differences in metal concentrations between land cover types and/or depths [40]. Significance was determined at the $\alpha < 0.05$ level. While analyses were performed on untransformed data, data presented in boxplots are natural log transformed due to the very large range of metal concentrations in the data set. All figures were created using the ggplot2 R package [41].

We also assessed the proportion of samples in the dataset with elevated metal concentrations using reported average concentrations in the A horizon of soils from across the United States from the USGS [42] (Table 1). We compared soil metal concentrations in the USGS AGBD2 to values from uncontaminated soils across the United States to assess how Alaskan soil metal concentrations compared relative to other biomes and climatic regions. Maps of sample locations showing the distribution of sampling sites, the land cover types from which the samples were collected, and the location of samples with elevated metal concentrations were created in ArcGIS (v. 10.7.1). We also mapped the locations of the sampling sites in the AGBD2 alongside locations of sites with known contamination from the Alaska Department of Environmental Conservation [43] to see if co-location with contaminated sites affected the concentrations we report.

## Results

### Sample distribution and metal concentrations across depth and land cover types

Sample locations in the dataset used for analysis were primarily in south and southwestern Alaska (Fig 1) and few sampling localities are in proximity to towns or settlements. The

**Table 1. Summary of metals concentrations.**

| Metal | Number of observations | Range (mg kg$^{-1}$) | Mean ± SD (mg kg$^{-1}$) | Median (mg kg$^{-1}$) | IQR (mg kg$^{-1}$) | US average (mg kg$^{-1}$)[a] |
|---|---|---|---|---|---|---|
| As | 961 | 0.39–14900 | 188 ± 1120 | 11.0 | 7.0–20.0 | 6.6 ± 19.6 |
| Cr | 1072 | 1.0–350 | 59.2 ± 38 | 56.0 | 33.3–78.0 | 37 ± 89 |
| Hg | 800 | 0.01–6090 | 30.4 ± 0.06 | 0.06 | 0.04–0.09 | 0.04 ± 0.17 |
| Ni | 1073 | 2.0–702 | 44.6 ± 61 | 28.3 | 19.0–41.6 | 18.5 ± 54.4 |
| Pb | 1081 | 2.0–720 | 21.1 ± 41.1 | 13.0 | 9.0–19.0 | 22.2 ± 46.6 |

IQR = interquartile range; first quartile–third quartile
[a]Values from Smith et al., 2013, Geochem. and Mineralogical Data for Soils of the Conterminous US, USGS Data Series 801

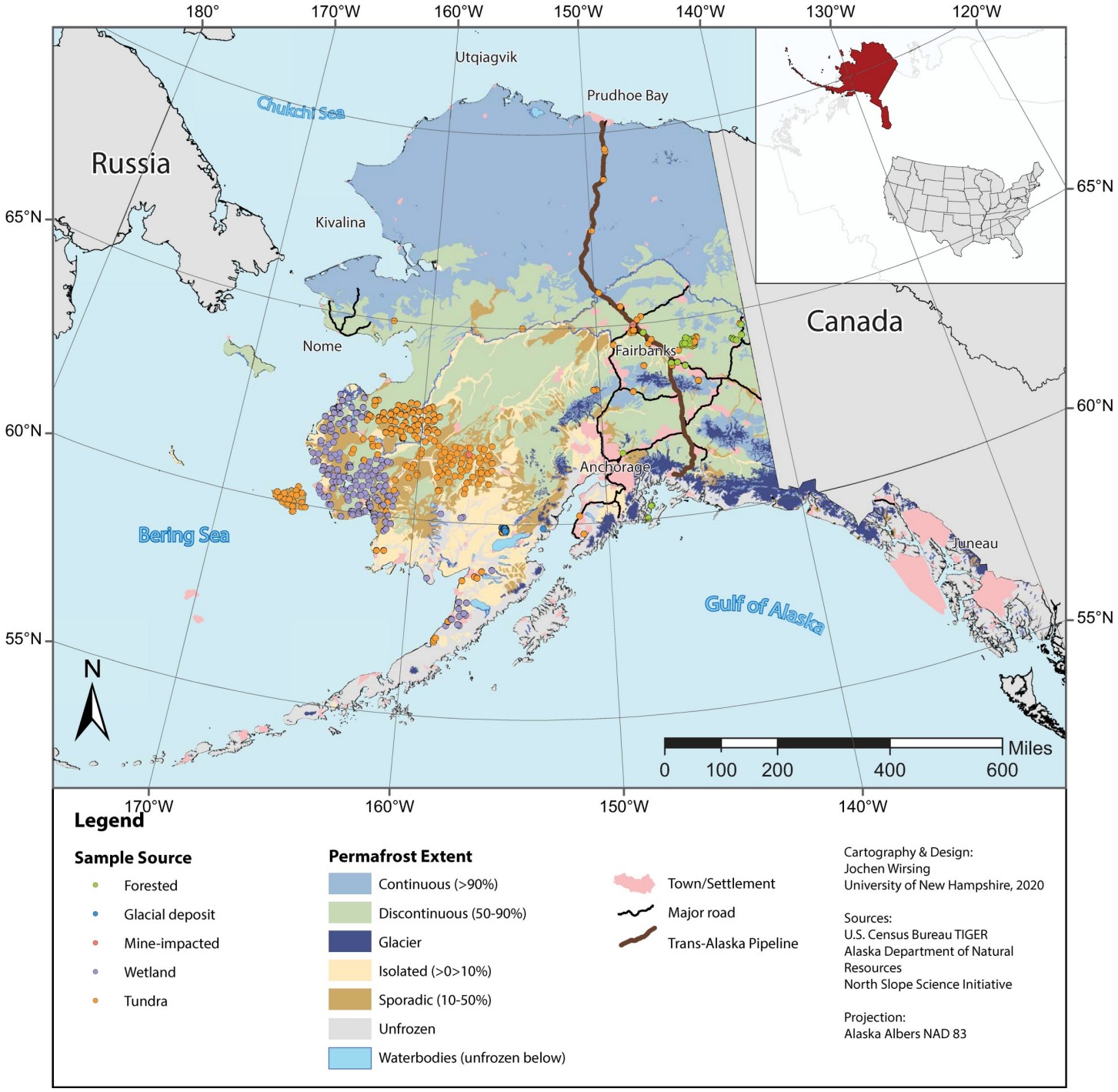

**Fig 1. Overview map of study area.** Map shows sampling locations, as well as towns and villages, major and minor highways, lakes, rivers, and glaciers. Sampling locations are color coded by land-cover type. Reprinted map layers from the US Census Bureau [44], Alaska Department of Natural Resources [45], and the North Slope Science Initiative for the US Fish and Wildlife Service [46].

dataset also contained very few (< 10) samples located in the continuous permafrost zone; the bulk of sample locations were located in discontinuous and sporadic permafrost. Metal concentrations are summarized in Table 1.

Metal concentrations varied between sampling depth intervals of 0–10 cm, 10–30 cm, and below 30 cm for all metals except As (Fig 2). While Ni concentrations are highest above 10 cm ($\chi^2_{[2]}$ = 63.38, p < 0.001; Kruskal-Wallis multiple comparisons of means and Fisher's Least Significant Difference tests for all comparisons reported here), Cr, Hg, and Pb have higher concentrations at depth. Concentrations are highest in the 10–30 cm depth interval for Cr ($\chi^2_{[2]}$ = 43.14, p < 0.001) and Pb ($\chi^2_{[2]}$ = 11.39, p = 0.003) and below 30 cm for Hg ($\chi^2_{[2]}$ = 6.68, p = 0.035). Metal concentrations also varied between land cover types (S1 Fig). Wetland sites had the lowest concentration of both As and Pb ($\chi^2_{[4]}$ = 99.76, p < 0.001; $\chi^2_{[4]}$ = 11.91, p < 0.001). Forested sites had higher concentrations of Cr and Ni and lower concentrations of Hg than other vegetated land cover types. Mine-impacted soils have higher concentrations of all metals except Ni, for which mine-impacted sites and forested sites had comparable concentrations (p < 0.001 for all).

## Comparison to US soil averages

Concentrations of As, Cr, Hg, Pb, and Ni in soils across Alaska exceed US average soil concentrations [42] (Fig 3 and Table 1). We quantified the percentage of samples with metal

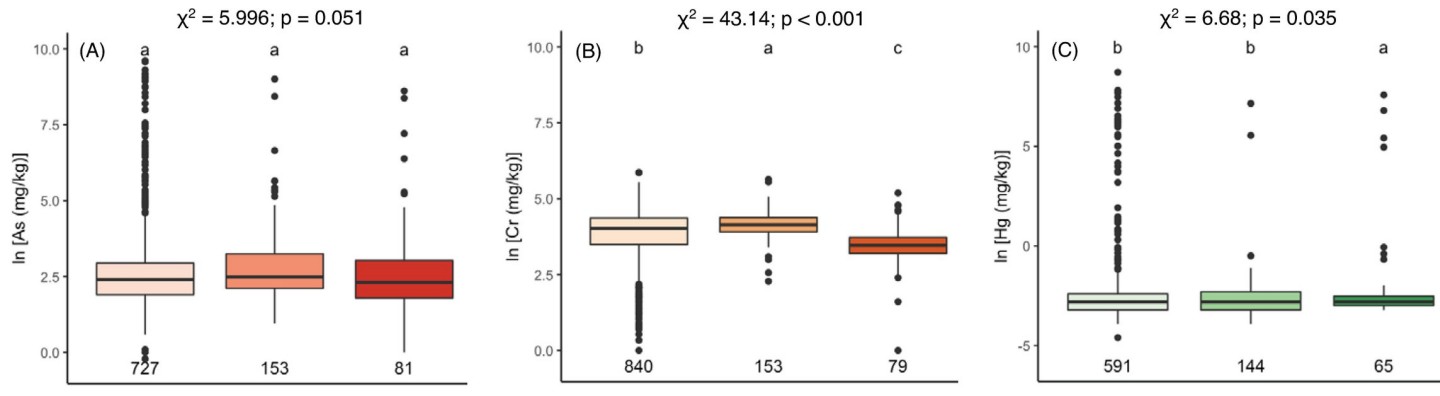

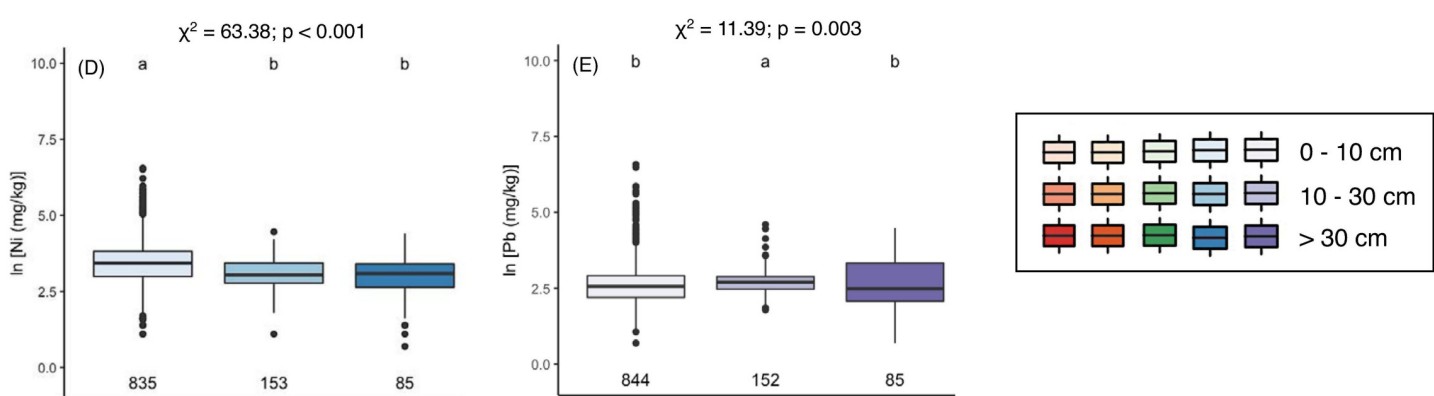

**Fig 2. Metal concentrations by sample depth.** Concentrations of As (A), Cr (B), Hg (C), Ni (D), and Pb (E) across binned sampling depths. All concentrations are displayed on a natural log scale. Chi-squared test statistic and alpha value results of the Kruskal-Wallis tested used to determine differences by depth for each metal are reported above each plot, DF = 2 for all. Letters above boxes represent groups determined by the Fisher's Least Significant Difference test. Numbers below boxes represent the number of observations from each source for each metal.

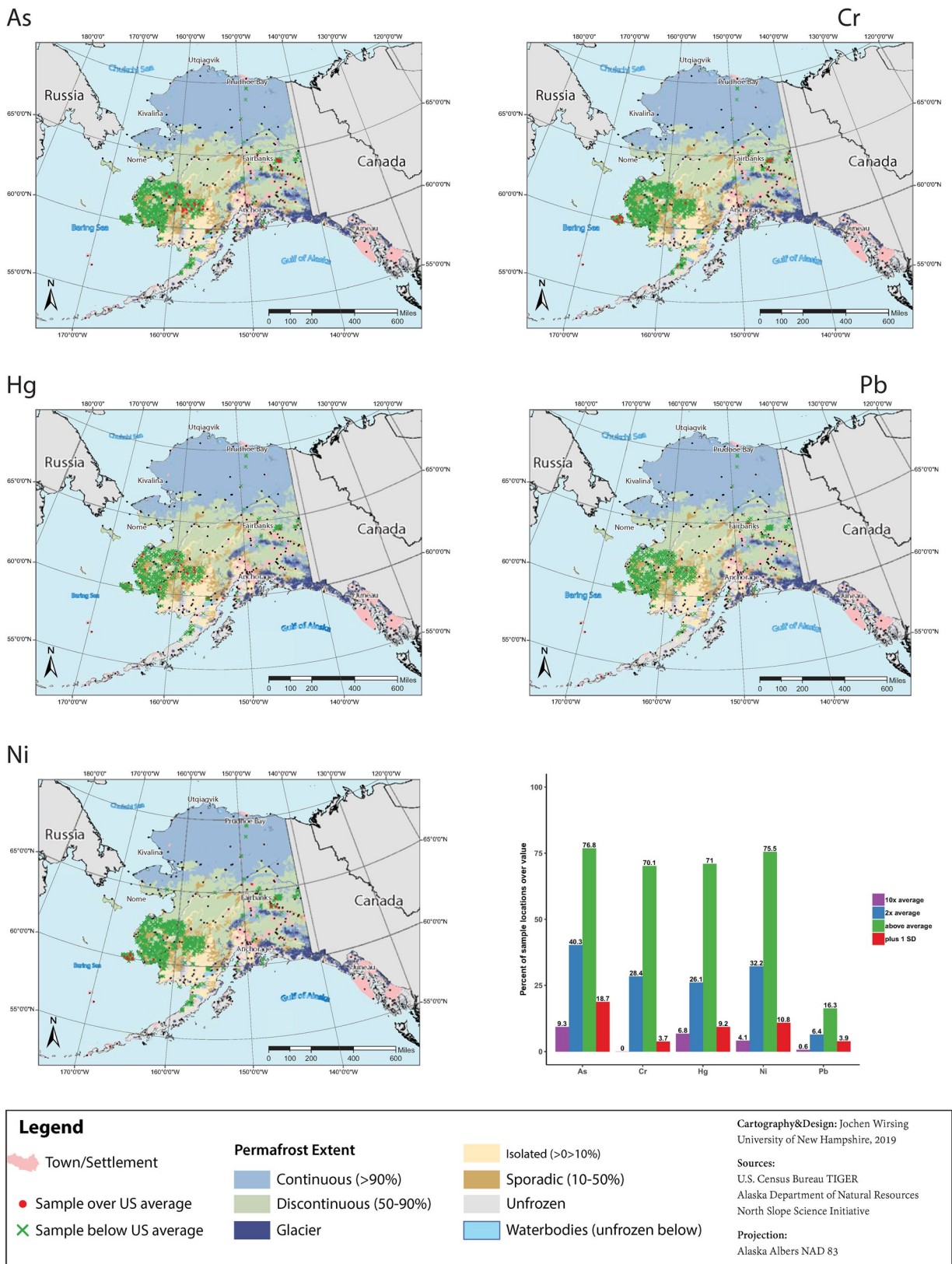

**Fig 3. Location and proportion of samples over US averages for soil metal concentrations.** Maps display sample locations with concentrations of As, Cr, Hg, Ni, Pb that do (red) or do not (green) exceed averages for US soils in Smith et al. (2013) by one standard deviation or more. Lower right panel shows the percentage of samples with metal concentrations that are greater than average values, as well as the proportion that exceed the US average by a factor of 2, by a factor of 10, and by more than one standard deviation of the average or more. The percentages above each bar are not cumulative. Reprinted map layers from the US Census Bureau [44], Alaska Department of Natural Resources [45], and the North Slope Science Initiative for the US Fish and Wildlife Service [46].

concentrations greater than reported US averages, as well as those that exceeded the reported averages by a factor of 2, a factor of 10, and by 1 standard deviation of the national average.

Over 70% of soil samples had concentrations of As, Cr, Hg, and Ni greater than published average values of soil metals concentrations across the United States. Further, 18% of samples had Hg concentrations higher than 0.1 mg kg$^{-1}$, compared to 12% found in a synthesis of soils in the western USA by Obrist et al. [47]. Over 25% of samples had concentrations that exceeded average concentrations of As, Cr, Hg, and Ni by a factor of 2, and nearly 10% of soil samples in the AGDB2 had As concentrations that exceeded the US average by a factor of 10. Lead concentrations were generally lower than the US average, and only 16.3% of samples reported here were greater than the reported US average concentration. Across the metals, 3.7% to 18.7% of the samples in the database had elevated concentrations that exceeded the US average values by one standard deviation or more (Fig 3), with As having the largest and Cr having the smallest proportion of samples with such elevated concentrations. Sampling locations in which soils exceed reported averages by one standard deviation or more are distributed throughout the study region (Fig 3).

## Discussion

An overarching challenge in synthesizing large datasets such as the AGDB2 are the large ranges and variances found in the data, as shown in Table 1. Despite the large variance in the data, the heavy metal concentrations in the soils in the AGDB2 dataset are within the range of those previously reported in other parts of the Arctic. For example, the median values we report for As, Ni, and Pb of 11.0, 28.3 and 13.0 mg kg$^{-1}$, respectively, fall within the range previously reported in permafrost affected soils from the Lena Delta in Siberia of As ranging from 1.29 to 11.3 mg kg$^{-1}$, Ni from 5.12 to 30.2 mg kg$^{-1}$, and Pb from 2.41 to 13.1 mg kg$^{-1}$ [13]. The interquartile range (IQR) we report for Ni and Pb is the same order of magnitude as the range of concentrations reported by Barker et al. [27] in soil from the Imnavait Creek watershed in Alaska of 13.9–23.7 mg kg$^{-1}$ and 4.8–13.1 mg kg$^{-1}$, respectively. Halbach et al. [48] measured a suite of metal concentrations in Svalbard, and they report median As, Cr, and Ni concentrations approximately 2 to 4 times lower than the median concentrations reported in this study. Mercury and Pb concentrations from Svalbard [48] of 0.023 to 0.107 mg kg$^{-1}$ and 10.6 to 10.7 mg kg$^{-1}$ are the same order of magnitude as our reported medians of 0.06 mg kg$^{-1}$ and 13.0 mg kg$^{-1}$. Similar levels of soil Hg have been reported from numerous sites across the Arctic, including the Canadian High Arctic [49], the Taimyr Peninsula [50], Lena Delta [13] in Siberia, Alaska [12], and Sweden [51]. The median and IQR of Hg concentrations reported here are in agreement with previous work which finds that Hg in Arctic soils tends to range from below detection (~0.01 mg kg$^{-1}$) to below 0.1 mg kg$^{-1}$.

To contextualize soil metal concentrations and their relevance to human health, we defined "elevated concentrations" as those greater than the average value for US soils by one standard deviation or more. Soil metals with elevated concentrations are distributed across the study area in Alaska regardless of land cover type and depth, and for the metals examined in this study, between 3.7% and 18.7% of samples had elevated concentrations (Fig 3). While we originally intended to contextualize our findings against human health guidelines, we decided not

to for two primary reasons. One, health guidelines are generally for oral and/or dermal exposure and may exceed concentrations at which metal liberation through hydrologic and biogeochemical cycles is a concern. Furthermore, the acceptable heavy metal concentrations in soils allowed by health guidelines vary widely across countries; for example, Kamunda et al. [52] report maximum allowable limits of the metals examined in this study in soils from different nations that vary over one to two orders of magnitude. Given the lack of international consensus amongst health professionals and the fact that these comparisons are highly sensitive to the standards selected, we chose to forgo a comparison to health standards. Instead, we chose to base our definition of "elevated concentrations" on observations of other soils.

Concentrations of As, Cr, Hg, Ni, and Pb in mine-impacted areas of Alaska were especially elevated (Fig 2), likely due to heavy metal air pollution from mines and subsequent wet/dry deposition onto soils [10,53]. However, soils with metals concentrations that exceed average values from the contiguous United States are distributed across the study region and are not isolated to mine-impacted sites or sites with close proximity to sources of anthropogenic contamination (S2 Fig). The proportion of samples with elevated concentrations were comparable when calculated with and without mine-impacted sites (S2 Fig), indicating that the small number (n < 20 for all metals, S1 Fig) of mine-impacted sites did not disproportionately impact the proportion of samples with elevated concentrations. This suggests that "pristine" soils in Alaska (i.e. those from outside areas with heavy anthropogenic disturbance) can also contain elevated levels of heavy metals which may be released through warming and thaw. Aside from anthropogenic contamination, heavy metals in soils can be derived from lithogenic pools that become distributed across surface environments through natural biogeochemical cycles and near-surface processes (i.e. erosion, weathering, pedogenesis) [54]. Furthermore, the bias towards surficial soils in the AGDB2 (Fig 2) impedes distinguishing if elevated concentrations are a result of background levels or from surface contamination.

Regardless of the source of heavy metals, their potential liberation as permafrost thaw and climatological drivers alter hydrological and biogeochemical cycles in Arctic and sub-Arctic regions is a pressing concern. Critical gaps in publicly available data for heavy metals in these soils preclude a thorough assessment of the pool of heavy metals that may be released from thawing permafrost. Over 99% of sample locations within AGDB2 are located in sporadic, isolated, and discontinuous permafrost, which combined make up only ~50% of Alaska [46]. Soil metal concentrations in the AGDB2 severely underrepresent soils in continuous permafrost. This limits our ability to infer whether soils in the continuous permafrost zone in Alaska sequester heavy metals. Likewise, 74% of the soil samples in AGDB2 were surface samples between 0–10 cm. Metal concentrations tended to be higher below 10 cm depth (Fig 2), suggesting that assessing metal concentrations in surface soil samples alone may not accurately reflect heavy metal concentrations through the soil profile. Most previously published work on heavy metals in permafrost-affected soil focuses on the seasonally thawed layer and/or surface soil, which further inhibits inference about the sequestration of heavy metals in permafrost. In two studies that did investigate deeper soil horizons, metals were found to be higher both just above the permafrost table [13] and within permafrost [27], a reflection that the prevalence of heavy metals in permafrost soil profiles may be complicated by site lithology and hydrology, mobility of the heavy metal in question, and previous thaw history. We suggest that future research on soil metal concentrations should include sampling of deeper soil, including permafrost.

Future work should attempt to constrain how heavy metals in soils move through terrestrial, aquatic, and human ecosystems in a warming Arctic, and the implications of widespread elevated soil metal concentrations to communities in the Alaskan Arctic. Heavy metals have long been found in human blood and tissues in Arctic communities [28], largely through

exposure to heavy metals through food [55]. Arctic communities are particularly susceptible to threats to water security from climate change [28], and potential liberation of metals through permafrost thaw may additionally threaten water quality in the Arctic [7,27]. Our study shows that high concentrations of heavy metals are distributed in soils across Alaska, suggesting permafrost thaw and associated changes in the environment may present an unaccounted-for pathway of heavy metal exposure. This conclusion is supported by previous observations in Alaska of suspended river and coastal sediments enriched in heavy metals [56,57], increased elemental and sediment loading into freshwater ecosystems as permafrost thaws [58,59], and of a seasonal signature of stream water metal concentrations that coincides with the depth of seasonal thaw [27]. Given recent observations of widespread and rapid permafrost thaw due to ongoing warming [60] and increases in wildfires that accelerate thermokarst expansion [61], more attention to the threat of heavy metal liberation from thawing permafrost is needed. A critical first step is ameliorating gaps in the existing data, which underrepresent continuous permafrost, deeper soils horizons, and locations near human communities.

## Conclusion

While the metal concentrations in both USGS datasets include large variances, our synthesis of publicly available data on soil concentrations of As, Cr, Hg, Ni, and Pb in Alaska indicates that metal concentrations in soils in southern and southwestern Alaska often exceed average US soil concentrations by a factor of 2 or more. The prevalence of As, Cr, Hg, Ni, and Pb in soils largely in the discontinuous and sporadic permafrost zone emphasizes the need for better understanding of how permafrost thaw alters the mobility and cycling of heavy metals. This synthesis highlights three gaps in publicly available data of heavy metal concentrations in permafrost-affected soils in Alaska: the lack of sampling near human populations, limited data on metal concentration in the continuous permafrost zone, and a bias in previous sampling towards surface (upper 10 cm) soils. Filling these data gaps will be necessary for understanding the potential for liberation of heavy metals from permafrost thaw into food and water resources that may present an unaccounted-for pathway of heavy metal exposure for communities in a warming Arctic.

## Supporting information

**S1 Fig. Metal concentrations by land cover type.** Concentrations of As (A), Cr (B), Hg (C), Ni (D), and Pb (E) across different land cover types. All concentrations are displayed on a natural log scale. Chi-squared test statistic and alpha value results of the Kruskal-Wallis tested used to determine differences by land cover type for each metal are reported above each plot, DF = 4 for all. Letters above boxes represent groups determined by the Fisher's Least Significant Difference test. Numbers below boxes represent the number of observations.
(EPS)

**S2 Fig. Location of known contaminated sites and impact of mine sites on the proportion of samples with elevated metal concentrations.** (Left) Overview map of study area including known contaminated sites from the Alaska Department of Environmental Conservation (https://dec.alaska.gov/Applications /SPAR/PublicMVC/CSP/Search/). (Right) Comparison of the percent of sampling sites in our cleaned dataset that had metal concentrations over US national averages by one standard deviation or more (upper right) and human health guidelines (upper left) with and without sites denoted in the dataset as mine-impacted included. Reprinted map layers from the US Census Bureau [44], Alaska Department of Natural Resources [45], and the North Slope Science Initiative for the US Fish and Wildlife Service [46].
(EPS)

**S1 File. Post-processing data from the USGS Alaskan Geochemical Database Version 2.0 used in our analysis.**
(CSV)

## Acknowledgments

We would like to thank the members of the New England Arctic Network (NEAN) for their feedback throughout the development of this work, especially NEAN members Jack Dibb, Katherine Duderstadt, Robyn Barbato, Ruth Varner, Julie Bryce, Florencia Fahnestock, and Stacey Doherty. We would also like to thank the Arctic Science IntegrAtion Quest (ASIAQ) for their support and the academic editor and 5 reviewers of this manuscript for their time and thoughtful feedback. Database searches, data analysis, and preliminary manuscript preparation was conducted through a course offered by the UNH Department of Natural Resources and the Environment, and we thank the department for their support of the course.

## Author Contributions

**Conceptualization:** Clarice R. Perryman, Jochen Wirsing, Kathryn A. Bennett, Owen Brennick, Apryl L. Perry, Nicole Williamson, Jessica G. Ernakovich.

**Data curation:** Clarice R. Perryman, Jochen Wirsing, Kathryn A. Bennett, Owen Brennick, Apryl L. Perry, Nicole Williamson, Jessica G. Ernakovich.

**Formal analysis:** Clarice R. Perryman, Jochen Wirsing.

**Software:** Clarice R. Perryman, Jochen Wirsing.

**Supervision:** Jessica G. Ernakovich.

**Visualization:** Clarice R. Perryman, Jochen Wirsing.

**Writing – original draft:** Clarice R. Perryman, Jochen Wirsing, Kathryn A. Bennett, Owen Brennick, Apryl L. Perry, Nicole Williamson, Jessica G. Ernakovich.

**Writing – review & editing:** Clarice R. Perryman, Jochen Wirsing, Jessica G. Ernakovich.

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
