## [Decision Letter · Decision Letter 0]

12 Dec 2019

PONE-D-19-30690

Heavy metal in the Arctic: Characterizing the distribution of five metals in soil in the Alaskan Arctic

PLOS ONE

Dear Ms. Perryman,

Thank you for submitting your manuscript to PLOS ONE. After careful consideration, we feel that it has merit but does not fully meet PLOS ONE’s publication criteria as it currently stands. Therefore, we invite you to submit a revised version of the manuscript that addresses the points raised during the review process.

We would appreciate receiving your revised manuscript by Jan 26 2020 11:59PM. To enhance the reproducibility of your results, we recommend that if applicable you deposit your laboratory protocols in protocols.io, where a protocol can be assigned its own identifier (DOI) such that it can be cited independently in the future. For instructions see: http://journals.plos.org/plosone/s/submission-guidelines#loc-laboratory-protocols

We look forward to receiving your revised manuscript.

Kind regards,

Yi Hu

Academic Editor

PLOS ONE

Journal Requirements:

2. We note that Figures 1, 3 and S2 in your submission contain map images which may be copyrighted.

a. You may seek permission from the original copyright holder of Figures 1, 3 and S2 to publish the content specifically under the CC BY 4.0 license. 

Reviewers' comments:

Reviewer's Responses to Questions

**Comments to the Author**

1. Is the manuscript technically sound, and do the data support the conclusions?

Reviewer #1: No

Reviewer #2: Yes

Reviewer #3: Yes

Reviewer #4: Yes

Reviewer #5: Yes

2. Has the statistical analysis been performed appropriately and rigorously? 

Reviewer #1: No

Reviewer #2: Yes

Reviewer #3: Yes

Reviewer #4: I Don't Know

Reviewer #5: Yes

3. Have the authors made all data underlying the findings in their manuscript fully available?

Reviewer #1: No

Reviewer #2: Yes

Reviewer #3: Yes

Reviewer #4: Yes

Reviewer #5: Yes

4. Is the manuscript presented in an intelligible fashion and written in standard English?

Reviewer #1: Yes

Reviewer #2: Yes

Reviewer #3: Yes

Reviewer #4: Yes

Reviewer #5: Yes

5. Review Comments to the Author

Reviewer #1: see attachement

The manuscript “Heavy metal in the Arctic” addresses important scientific problem and uses unique set of data which should be presented to the scientific community

However the treatment of the data, comparison with other available studies and discussion of mechanisms and consequences are not at the level suitable for an academic publication

Reviewer #2: The manuscript presents relevant new information and has high quality. However, the manuscript presents few points that need to be revised. The detailed comments are listed below:

Keywords

1. Delete all keywords already in title.

Introduction

1. Line 114 � Delete “to”.

Materials and Methods

1. Line 128: Delete “Alaskan Geochemical Database Version 2.0”.

Results

1. Table 1 presentation should be improved.

Discussion

1. Line 243 and all manuscript: mg/kg � mg kg-1.

2. Second paragraph: What implications can these results have on the environment?

Transfer to surface and subsurface waters?

Line 310: It does not reduce SOM's ability to retain metals, it only reduces its soil contents. Rewrite.

Reviewer #3: PLOS ONE

Manuscript Number: PONE-D-19-30690

Title: Heavy metal in the Arctic: Characterizing the distribution of five metals in soil in the Alaskan Arctic

The article is clearly presented and straightforward. While the authors did not synthesize any of their own data, they do present a large publically available dataset in a concise way that has not been done before for Alaska. Particularly, the key finding that there is a bias in sampling towards the upper 10 cm of soils and that may be biasing values to be preferentially lower for some metals versus others that tend to accumulate more in the deeper subsurface is an important finding.

Overall, comments are relatively minor. There are two major edits that need to be addressed: (1) the authors need to be more clear when comparing Alaska values to ‘US averages’ and ‘health standards’, it is confusing and somewhat misleading as written (explained further below in the line items) and (2) the authors need to clarify background values for the different classification ‘bin’ types – there should be published values on background concentrations for Alaska soils (also would be important to compare to crustal abundances for the metals studied).

Line items:

Title: Is there a reason the first ‘metal’ mentioned is not plural – i.e. ‘Heavy metals in the Arctic’ – sounds odd as it is since you are studying more than one metal.

Lines 62-64: I would clarify that there are also heavy metals present in the Arctic naturally in the soils at low concentrations in addition to anthropogenic sources.

Line 83: consider changing ‘peoples’ to ‘communities’

Lines 86 and 90: consider changing ‘country’ to subsistence-derived’

Line 95: is there a verb missing between ‘and’ and ‘substantial’?

Lines 150-153: Can you go more in depth on the classification systems that you re-coded? For the re-coded ‘tundra’ category does the included ‘shrub-covered’ and ‘grassland/grazing’ mean there were underlain with permafrost? Go more in-depth about the original USGS land-cover categories and what they meant and why you chose to include those groups together. I would assume the ‘tundra’ re-coding includes only places with permafrost? As it is written, it is unclear what those categories actually mean.

Line 165: What do you mean by ‘there are no conclusive U.S. standards’… for what? Drinking water? Soils? Fish? The US has drinking water standards, soil standards, etc. What do you mean by that statement?

Lines 171-172: Please discuss the criteria for how ATSDR determined the average concentrations of these metals in soils. It will help with your later comparisons for Alaska. Was Alaska included in the ATSDR calculations?

Figure 2: resolution looks blurry – it looks different than the other images in terms of resolution

Line 223: Which human health standard are you referring to? This whole paragraph refers to human health standards and US averages – list those standards and averages for referring to

Figure 3: Alaska is known to have high arsenic values naturally in their soils (in addition to mining activities) – I think it’s important to note the background values for all these metals. Arsenic in Alaska, for example, is always going to have higher than US average soils because there is an abundance of arsenic in soils up here – not necessarily due to ‘contamination’ – that is known. There needs to be a discussion of background values and crustal abundances for all the metals examined here. Also, please include what ‘health limit’ you are referring to on the figure somewhere – either in the caption or on the plot. The value and also what type of limit – drinking water, etc. It is very unclear, as is.

Methods section: you mentioned that you separated published values based on analytical method – any information on that in the document? Are some analyses different than others? There should be some mention of that in the results, etc.

Section ‘Comparison to health standards and US averages’: Does the US average include Alaska and Hawaii samples or not?

Discussion, 1st paragraph: Please note somewhere within this text that you are comparing average values for the entire state of Alaska (including legacy and active mining sites and likely some military sites) with concentrations from pristine sites (citations 12, 13, 45, 46) – the values should be different so that is not a new finding. The paragraph is a bit misleading as written – what would be more appropriate is to compare areas in Alaska with no known anthropogenic sources (Hg will be difficult to do this with) to the citations 12, 13, 45, and 46 and vice versa – compare legacy contamination with the ‘high’ values found in this report. Or only look at the ‘arctic’ values where there is little contamination.

Line 262: except Hg? You mentioned earlier that Hg was higher in the 0-10 cm bin.

Lines 268-271: Remove or reword the statement about permafrost acting as a barrier to metal migration – this statement as written assumes the metal concentrations found in the two studies mentioned (Antcibor and Barker) migrated down into permafrost and that is why the concentrations are higher. Some of these soils are very old and are a result of glacial deposits or wind-blown deposition processes over time and not as a result of metals leaching downward from the surface.

Line 291: Include citation for arsenic mobility, same for the next line on chromium

Reviewer #4: My review is submitted as an attachment. I suggest major changes in the manuscript and request that the authors consider revising some of their health standards and US soil average values. I do believe that the paper should be published after changes submitted.

Reviewer #5: Summary

Heavy metals in the environment can result in detrimental effects on the health of the public. Metals have been deposited across the Alaskan Arctic over time, becoming sequestered in permafrost. Melting permafrost due to climate change could result in the release of these metals to the environment and increase exposure to humans and wildlife. This study evaluated soils concentrations of heavy metals across the Alaskan Arctic using publicly available data. Soil heavy metal concentrations in the Alaskan Arctic were generally found to be elevated compared the average concentrations of the contiguous US. This study provides a novel synthesis of soil heavy metal concentrations across the Alaskan Arctic and identifies important gaps for future research efforts. Additionally, this study improves the understanding of health risks associated with possible heavy metal releases from permafrost thaw in the Alaskan Arctic.

Major comments

It’s difficult to determine from the methods, but were concentrations of all 5 heavy metal known for any/some/many of the included sampling locations? This could be made clearer in the method. While it may be outside the scope of this study, it would be interesting to look at the relationship among the metals sampled at the same location (if this occurs).

Minor comments

Line 47: Change “concentration” to “concentrations”

Line 76 (88, 113): Be consistent in including the element symbols in the first mention and using the symbol in subsequent mentions

Line 101: It would be good to focus on health effects that may result from exposure due to release from permafrost, it does not seem relevant to include “respiratory problems from inhalation”

Line 111: The term “permafrost-affected soils” seems strange, maybe use different terminology?

Line 164 -170 and 216: It would be useful to include how you determined the health standards based on types of exposure. For instance, you reference oral and dermal exposure in line 216 for As. This information should be included in the methods, did you use oral and dermal health standards are each metal used? Were other types for exposure standards used?

Line 170-175: Include that these averages are for uncontaminated soils (as explained in line 278)

Line 198 and 208: Change “sample sources” to “land-cover types”, be sure to keep the terminology consistent

Line 212: Change “source” to “type”

Line 298-299: Remove “have been found”

Lines 321-328: It would be good to include some discussion about the depth profile (> 10, 10-30, <30 cm) and how that might relate or be important to transport from melting permafrost

Line 334: change “concentration” to “concentrations”

Fig 1. The pink color of the mine-impacted sites is hard to see, particularly due to the similarity to the pink color of the town/settlements

Fig. 3. This is figure is blurry

Fig. 3. There are some orange dots in the Ni map, are those supposed to be there?

6. PLOS authors have the option to publish the peer review history of their article (what does this mean?). If published, this will include your full peer review and any attached files.

Reviewer #1: No

Reviewer #2: No

Reviewer #3: No

Reviewer #4: No

Reviewer #5: No

---

## [Author Response · Author response to Decision Letter 0]

14 Mar 2020

Our responses to all Reviewer comments can be found in our Response to Reviewers document. A summary of the revisions and improvements is also provided in our Cover Letter for the revised manuscript.

---

## [Decision Letter · Decision Letter 1]

3 Apr 2020

PONE-D-19-30690R1

Heavy metals in the Arctic: Distribution and enrichment of five metals in Alaskan soils

PLOS ONE

Dear Ms. Perryman,

Thank you for submitting your manuscript to PLOS ONE. After careful consideration, we feel that it has merit but does not fully meet PLOS ONE’s publication criteria as it currently stands. Therefore, we invite you to submit a revised version of the manuscript that addresses the points raised during the review process.

We would appreciate receiving your revised manuscript by May 18 2020 11:59PM. To enhance the reproducibility of your results, we recommend that if applicable you deposit your laboratory protocols in protocols.io, where a protocol can be assigned its own identifier (DOI) such that it can be cited independently in the future. For instructions see: http://journals.plos.org/plosone/s/submission-guidelines#loc-laboratory-protocols

We look forward to receiving your revised manuscript.

Kind regards,

Yi Hu

Academic Editor

PLOS ONE

Reviewers' comments:

Reviewer's Responses to Questions

**Comments to the Author**

1. If the authors have adequately addressed your comments raised in a previous round of review and you feel that this manuscript is now acceptable for publication, you may indicate that here to bypass the “Comments to the Author” section, enter your conflict of interest statement in the “Confidential to Editor” section, and submit your "Accept" recommendation.

Reviewer #1: All comments have been addressed

Reviewer #2: All comments have been addressed

Reviewer #3: All comments have been addressed

Reviewer #4: (No Response)

Reviewer #5: All comments have been addressed

2. Is the manuscript technically sound, and do the data support the conclusions?

Reviewer #1: Yes

Reviewer #2: Yes

Reviewer #3: Yes

Reviewer #4: Yes

Reviewer #5: Yes

3. Has the statistical analysis been performed appropriately and rigorously? 

Reviewer #1: Yes

Reviewer #2: Yes

Reviewer #3: Yes

Reviewer #4: Yes

Reviewer #5: Yes

4. Have the authors made all data underlying the findings in their manuscript fully available?

Reviewer #1: Yes

Reviewer #2: Yes

Reviewer #3: Yes

Reviewer #4: Yes

Reviewer #5: Yes

5. Is the manuscript presented in an intelligible fashion and written in standard English?

Reviewer #1: Yes

Reviewer #2: Yes

Reviewer #3: Yes

Reviewer #4: Yes

Reviewer #5: Yes

6. Review Comments to the Author

Reviewer #1: The comments of 1st reviewer were adequately adressed, and the arguments of authors (where they disagree) are well taken. The paper is in the shape which is suitable for publication; it represents clear added value.

Reviewer #2: (No Response)

Reviewer #3: (No Response)

Reviewer #4: The revised manuscript (PONE-D-19-30690R1) is improved due to efforts by the authors to address numerous suggestions from the reviewers. An overreaching difficulty in writing the paper was trying to work with very large ranges and standard deviations (SD) in both the Alaska and the US soil data. I encourage the authors to mention this challenge in the beginning of the Discussion and again in the Conclusions.

I have a few suggestions listed below in order of appearance in the text.

1. Line 40 and many others. I do not think that various forms of the word “liberate” (e.g., lines, 40, 64, 69, 72 and several more throughout the text) best explain the process of “release” or “mobilization” of metals from previously frozen soils. I prefer not to see the much overused word “liberate” in the manuscript.

2. Line 41. “exacerbate contamination” implies that the soils are already contaminated. Although this is certainly true in a few cases, it does not seem to be the norm nor is it adequately proven. Therefore, “… potentially contaminating food and water” … seems more defendable.

3. Lines 46-57. After listing Items 1), 2) and 3), I prefer to see responses to each to follow in the same order. Therefore, the first sentence after the list would be “We found substantial clustering …”.

4. Overall, the Abstract, Introduction and Discussion could use some careful editing with a few hours of help from each of the other authors.

5. Table 1. Better use of significant figures is essential. Use of more than 3 significant figures seems unacceptable (e.g., 14900.0 [6 sig figs] goes to 14900 [3 sig figs]; 112.73 [5 sig figs] goes to 113 [3 sig figs]).

Pb (mean ± SD) is much more realistic at 21 ± 41, and so on.

Table 1 really prompted my introductory comment about challenges of the large ranges and SDs. Care is certainly required when comparing two data sets with such large ranges and SDs.

To me, range seems better than maximum and minimum (e.g., 2‒720 for Pb).

As an aside, in rethinking the statistics, I probably would have used medians and median

absolute deviations (MAD, as in Smith et al., 2013). No need for the authors to make this change.

6. Line 257 and many more. I previously commented “I see too much use of the word ‘comparable’. All data are comparable. The point is whether means for one set of data are statistically different or not from another set of data.”

I still see "comparable" used too often (e.g., lines 257, 259, 264, 267)

7. I think the authors make their point and thus the Conclusion seems to work well.

Reviewer #5: (No Response)

7. PLOS authors have the option to publish the peer review history of their article (what does this mean?). If published, this will include your full peer review and any attached files.

Reviewer #1: Yes: Oleg S. Pokrovsky

Reviewer #2: Yes: Tadeu Luis Tiecher

Reviewer #3: No

Reviewer #4: No

Reviewer #5: No

---

## [Author Response · Author response to Decision Letter 1]

30 Apr 2020

Our response to all reviewer comments can be found in our uploaded Response to Reviewers document. A summary of changes in this revision is also included in our Cover Letter for the resubmission.

---

## [Decision Letter · Decision Letter 2]

4 May 2020

Heavy metals in the Arctic: Distribution and enrichment of five metals in Alaskan soils

PONE-D-19-30690R2

Dear Dr. Perryman,

We are pleased to inform you that your manuscript has been judged scientifically suitable for publication and will be formally accepted for publication once it complies with all outstanding technical requirements.

With kind regards,

Yi Hu

Academic Editor

PLOS ONE

Additional Editor Comments (optional):

Reviewers' comments:

Reviewer's Responses to Questions

**Comments to the Author**

1. If the authors have adequately addressed your comments raised in a previous round of review and you feel that this manuscript is now acceptable for publication, you may indicate that here to bypass the “Comments to the Author” section, enter your conflict of interest statement in the “Confidential to Editor” section, and submit your "Accept" recommendation.

Reviewer #4: All comments have been addressed

2. Is the manuscript technically sound, and do the data support the conclusions?

Reviewer #4: Yes

3. Has the statistical analysis been performed appropriately and rigorously? 

Reviewer #4: Yes

4. Have the authors made all data underlying the findings in their manuscript fully available?

Reviewer #4: Yes

5. Is the manuscript presented in an intelligible fashion and written in standard English?

Reviewer #4: Yes

6. Review Comments to the Author

Reviewer #4: Great persistence on your part produced a much improved, interesting and useful manuscript.

Congratulations.

7. PLOS authors have the option to publish the peer review history of their article (what does this mean?). If published, this will include your full peer review and any attached files.

Reviewer #4: No

---

## [Editor Report · Acceptance letter]

8 May 2020

PONE-D-19-30690R2 

Heavy metals in the Arctic: Distribution and enrichment of five metals in Alaskan soils 

Dear Dr. Perryman:

I am pleased to inform you that your manuscript has been deemed suitable for publication in PLOS ONE. Congratulations! Your manuscript is now with our production department. 

With kind regards,

on behalf of

Prof. Yi Hu 

Academic Editor

PLOS ONE